

# Green city logistics path planning and design based on genetic algorithm

Limin Ran[1], Shengnan Ran[2] and Chunmei Meng[3]

[1] Flight College, Anyang Institute of Technology, Anyang, Henan, China
[2] Department of Physical Training, Leshan Normal University, Leshan, Sichuan, China
[3] College of Marxism, Kunming University of Science and Technology, Kunming, Yunnan, China

## ABSTRACT

Effective logistics distribution paths are crucial in enhancing the fundamental competitiveness of an enterprise. This research introduces the genetic algorithm for logistics routing to address pertinent research issues, such as suboptimal scheduling of time-sensitive orders and reverse distribution of goods. It proposes an enhanced scheme integrating the Metropolis criterion. To address the limited local search ability of the genetic algorithm, this study combines the simulated annealing algorithm's powerful local optimization capability with the genetic algorithm, thereby developing a genetic algorithm with the Metropolis criterion. The proposed method preserves the optimal chromosome in each generation population and accepts inferior chromosomes with a certain probability, thereby enhancing the likelihood of finding an optimal local solution and achieving global optimization. A comparative study is conducted with the Ant Colony Optimization, Artificial Bee Colony, and Particle Swarm Optimization algorithms, and empirical findings demonstrate that the proposed genetic algorithm effectively achieves excellent results over these algorithms.

## INTRODUCTION

Distribution is a crucial aspect of modern logistics, accounting for more than half of logistics service costs (*Chen, Ma & Sun, 2019*; *Duan, 2018*). Scientific route planning can enhance the level of service in a city, while also reducing costs. Currently, logistics management involves optimizing distribution routes to improve performance, enhance customer experience and reduce business costs (*Meng & Li, 2020*; *Zhizhong, 2018*). Furthermore, optimizing logistics distribution paths can indirectly contribute to solving social problems such as emission pollution and road congestion, promoting the sustainable development of society in all aspects by achieving internal unity of national resources, environment, and values. This is why logistics path planning optimization schemes have become the focus of research for experts, scholars, consulting agencies, and related enterprises. The problem of vehicle routing has received significant attention from scholars and experts in logistics science, operations research, combinatorial mathematics, network planning, and other fields (*Hu, Jiang & Zhou, 2020*; *Dantzig & Ramser, 1959*; *HuZhi-Hua, 2011*). In

Corresponding author
Limin Ran, 20160866@ayit.edu.cn

recent decades, the research results of experts in various fields have been dynamic, with many problem scenarios and models being expanded, and new solution methods emerging

*Venkatachalam & Sundar (2016)* presented distance and cost as salient factors and utilized an intelligent optimization algorithm to establish a path-planning model. The model is decomposed by splitting the objective and subsequently adding the shortest routes, solved by each substructure, to achieve the optimal solution (*Baum et al., 2016*). *Fukasawa, Qie & Yongjia (2014)* discovered that the various turns on the route impair the quality of different models in the vehicle distribution process, and therefore used the mixed integer programming method to solve the problem. *Mahmoudzadeh, Powers & Sammut (2015)* formulated diverse techniques to minimize the risk of underwater vehicles. Based on the general Vehicle Routing Problem (VRP), *Xiang, Hao & Zhang (2021)* introduced an open, multi-vehicle, and customer random demand vehicle routing problem and proposed an adaptive two-stage heuristic algorithm. The efficacy of Xiang's method was demonstrated through four rigorous practical tests. *Sun (2021)* constructed a VRP problem model with a vehicle maximum load constraint and employed an adaptive strategy to enhance the evolutionary process of the genetic algorithm. The number of distribution vehicles and the transportation capacity were significantly improved, thus ensuring the satisfaction of customer demand. However, the above solutions exhibit intricate scenarios that need to be resolved, such as logistics and distribution of high-value loss compensation goods, reverse distribution of returns, and distribution of low-performance time effect orders.

Due to the simple design and single operation mode of each genetic operator, the traditional genetic algorithm is harsh on the setting of population size and parameters. If the parameter setting is not reasonable, it is easy to fall into local optimal, and the solving accuracy is not high when the fitness of each chromosome is relatively close. The Metropolis criterion of simulated annealing algorithm can accept or reject the new solution with certain probability, so as to improve the defect of genetic algorithm which is easy to fall into local optimal. The combination of the two can solve complex routing problems better. Therefore, we propose a GA-based LPP scheme that introduces the Metropolis criterion (ULDGA), and verify the validity and reliability of the model and design algorithm through experiments.

Section 'Related Works' summarizes the vehicle routing problem, the section 'The ULDGA Scheme' introduces the modeling idea of ULDGA, and section 'The analysis of the simulation and result' simulates the model and analyzes the results.

## RELATED WORKS

Since the advent of the Vehicle Routing Problem (VRP), the problem has been explored in various forms. *Dantzig & Ramser (1959)* conceptualized the logistics routing problem as a system in which one or more distribution centers provide logistics services to multiple customers under certain constraints and optimize the objective function to complete all distribution tasks. *Wang & Bai (2022)* proposed the segmentation of the VRP set, optimized the VRP based on this segmentation, and established the simplest VRP model, which laid the foundation for subsequent VRP research. The Constrained Vehicle Routing Problem

(CVRP) typically only involves constraints on vehicle load. This model has been studied for the longest period and boasts significant achievements. Various algorithms, including both exact and heuristic algorithms, have been proposed to address the CVRP, including the Vehicle Routing Problem with Time Windows (VRPTW). The VRPTW is the most widely studied model, and many other models have been derived from it. It incorporates time window constraints based on the CVRP. *Solomon (1986)* noted that the addition of a time window to the VRPTW significantly increases the complexity of the solution. Currently, there are numerous intelligent optimization algorithms available to solve such problems, including Pick-up and Delivery VRP (VRPPD), in which customers require both pick-up and delivery services. The problem of distributed distribution VRP is one such problem, and it is more complex than regular VRP because the distribution center is usually assigned to different customers, and then the single distribution center method is used to solve the problem.

In the realm of multi-model vehicle routing problems (VRP), the entirety of the distribution task necessitates the utilization of diverse vehicular types, each of which may differ in number and cost. This particular quandary differs from the open VRP (OVRP) as proposed by *Guo, Shen & Zhu (2014)*. In contrast to the VRP, the OVRP does not mandate the return of the vehicle to the distribution center following the task's completion, and the final distribution route is not necessarily straightforward. In the modern era, scholars have endeavored to address these quandaries. For instance, *Xu, Qi & Fen (2020)* introduced the C-W savings algorithm, while *Mei & Zhou (2018)* proposed a scanning algorithm to overcome the problem of unlimited vehicle numbers in VRP. *Liu & Tang (2022)* developed a time-difference insertion heuristic algorithm inspired by the classical insertion heuristic algorithm of the VRPTW (*Mei & Zhou, 2018*), which was also analyzed for its complexity. In the same year, *Xiang, Hao & Zhang (2021)* proposed a solution for the TDVRP by extracting Vienna's street traffic information data and simulating the path using Dijkstra and variable neighborhood search algorithms, which yielded promising results. *Mungwattana et al. (2016)* established a dual-objective vehicle routing problem model based on the number of vehicles and total travel time. In light of the dynamic nature of customer needs, *Chen, Chen & Gao (2017)* tackled the dynamic demand vehicle routing problem with time windows, utilizing a hybrid algorithm that combines the harmony search and variable neighborhood descent algorithm. This hybrid algorithm was able to achieve high global search capability, thanks to its incorporation of both harmony search and rough neighborhood descent.

## THE ULDGA SCHEME

The idea of ULDGA is introduced in this chapter, which considers the constraint of a hard time window. VRPTW denotes several considerations of the time window requirement of customer points based on CVRP. The time window represents the time range allowed by the customer; that is, the customer has the deadline and the earliest start time, which can be divided as hard time constraint and soft time window constraint. Hard time windows require delivery vehicles to deliver services within customer-specified time ranges, and the solution beyond this time range is not feasible. While soft time window constraint: a soft

time window mandates punishing tasks that cannot be completed within the required time and forcing vehicles to pay waiting or delay fees for arriving early or late. Each customer's time window is known and does not change.

A mathematical model of ULDGA is also derived based on the number of vehicles and the length of the entire path, where function (1)'s goal is being able to reduce the cost of different cars.

$$\min z_2 = \sum_{k \in K}^{K} x_{ijk_{(i=0)}}. \tag{1}$$

Based on the above, we make this assumption that in addition to starting from the distribution center at the specified time, all vehicles need to return to the distribution center within the specified period. Figure 1 shows an example of ULDGA. The Metropolis criterion is an effective focus sampling method. When the system changes from one energy state to another, the corresponding energy changes from 1E to 2E, if 2E lt 1E, the system accepts this state; At the same time, considering carbon emission and nonlinear energy consumption factors, the path, power, time and load constraint are established. Otherwise, the system will gradually become stable after a certain number of iterations. Metropolis criteria are often expressed as:

$$p(1 \rightarrow 2) = \begin{cases} 1, & 2E < 1E \\ \exp\left(2E - \dfrac{1E}{T}\right), & 2E > 1E. \end{cases} \tag{2}$$

Metropolis criterion comprehensively considers the above factors in the genetic algorithm. The individual is accepted when an individual's fitness is higher than the population's average fitness. When an individual's fitness is not higher than the average fitness of the people, the Metropolis criterion is used to decide whether to accept. According to the Metropolis criterion, the particles at temperature $T$. The probability of equilibrium is $\exp\left(2E - \frac{1E}{T}\right)$. In which $E$ is the change in internal energy, using energy $E$. Simulation objective function value $f$ and temperature $T$. Simulation control parameters, that is, to get the solution combination for correlation test evaluation index. To satisfy $n$, according to customer's needs, the pure electric animal flow vehicles will start from the distribution center with full power, providing freight distribution and delivery tasks. In the process of carrying out the distribution task, the quantity of goods required by the default distribution center is 0, and the customer demand $i$; the quantity of goods needed is $q_i$.

As shown in Fig. 2, the model proposed in this article aims to minimize the total distribution cost under the premise of meeting all customer needs. When a pure electric logistics vehicle distributes goods, its total distribution cost includes the following five parts: fixed cost, driving cost, charging cost, penalty cost and carbon emission cost. The model uses the minimum total distribution cost as the optimization objective. In addition, the path planning model of pure electric vehicles considering carbon emission and nonlinear energy consumption is also constructed. First, according to the analysis of the influencing factors of electric vehicle power consumption, we know that the electric power consumption in the driving process is nonlinear. By analyzing the conversion process of mechanical

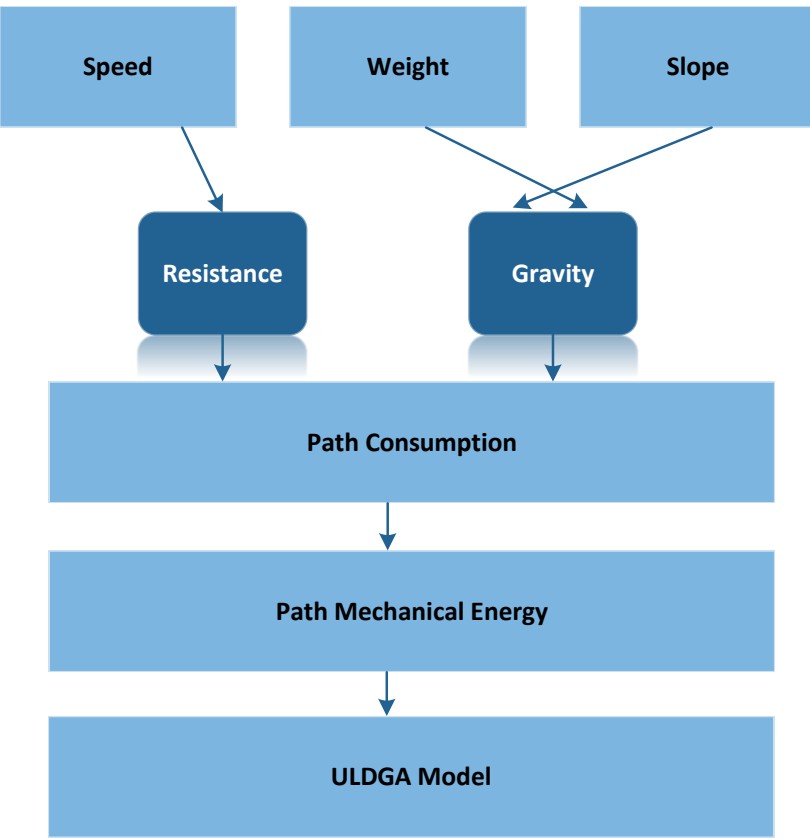

**Figure 1** **The modeling ideas of ULDGA.** We make the assumption that in addition to starting from the distribution center at the specified time, all vehicles need to return to the distribution center within the specified period.

energy, electrical energy and battery energy, the electric vehicle power consumption model is constructed. Secondly, the carbon emission is calculated based on nonlinear electricity consumption. We develop an optimization model for electrical vehicle distribution paths in which the distribution path, electricity, time, and load all play a role as constraints, while the optimization objective is to achieve the least total distribution cost possible.

Finally, because the original data cannot be directly processed in genetic algorithm, the distribution path needs to be transformed into chromosome path, that is, the path coding. Also, the coding methods of chromosome generally include binary coding, real coding, floating-point coding and symbol coding. In this article, natural number coding is adopted. In this article, several natural numbers are randomly arranged in each chromosome to solve the path-planning problem for electric vehicles, which can be divided into several parts. The number on each gene bit represents the number of demand points that have not been delivered, and the sequence of each gene bit represents the order of delivery vehicles visiting customer demand points. The chromosome length is $o + c + g$, Where $o$ means the distribution center, $c$ denotes customers, as well as $g$, is the charging stations. Set the distribution center number to 0. The customer number is set to $1, 2, 3, 4, 5, \ldots, n$. The

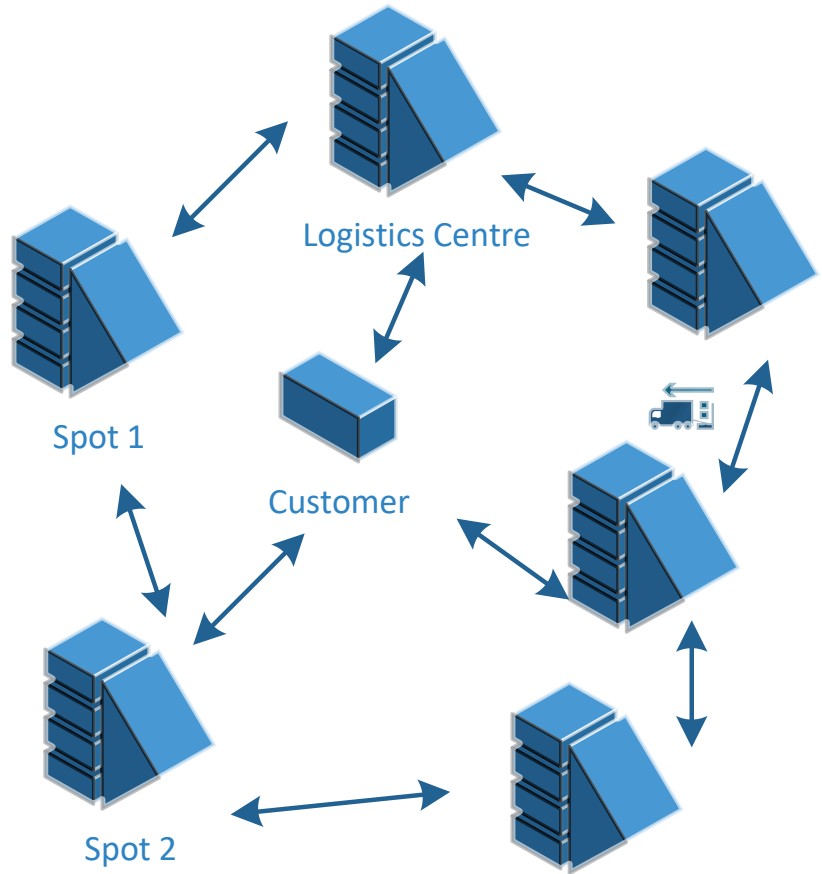

**Figure 2** **An example of ULDGA.** The proposed model aims to minimize the total distribution cost under the premise of meeting all customer needs.

number of the charging station is $n+1, n+2, n+3, n+4, n+m$. We insert the distribution center code 0 at the starting and end points, respectively, which constitutes a chromosome.

If two electric logistics vehicles are used to start from the distribution center o, the first vehicle will directly return to the distribution center after passing the customer's points 4,5,2 and 3, and the second vehicle will pass the customer's points 6 and 8, then arrive at the charging station 9 for charging, then arrive at the customer's point 1, and finally return to the distribution center. So the chromosome sequence is 0,4,5,2,3,0,6,1,3,2,0.

Figure 3 shows the algorithm process of ULDGA. The specific algorithm is as follows:

(a) Initial population, generate individual determine the mutation probability $p_0$.

(b) Start coding, and determine the natural number coding.

(c) To optimize and design the appropriate response function fitness$(l_i) = l^*/l_i$. In which fitness$(l_i)$ denotes the $i$ Individual fitness. There are two path lengths in the initial population: one represents the trajectory of the optimal chromosome ($l^*$), and the other means the trajectory of the current chromosome. Represents the path length of the optimal chromosome in the initial population ($l_i$). The fitness function calculates

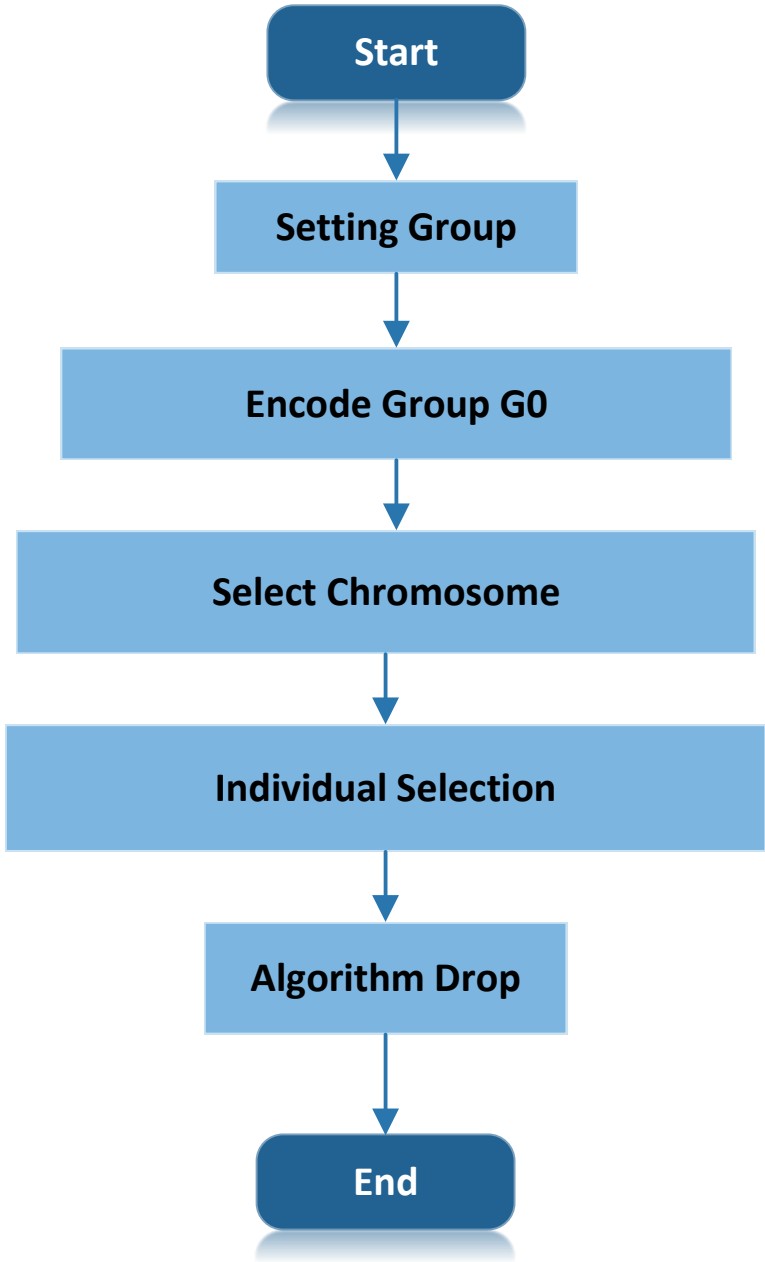

**Figure 3 ULDGA algorithm process.** The specific algorithm is as follows: initial population, generate individual determine the mutation probability. Start coding, and determine the natural number coding.

the individual fitness value, and the order is from small to large. The fitness function selected excellent chromosomes.

(d) Individual selection, the fitness function selection of keeping the best chromosome into the next generation.

(e) Cooling, choosing $T = aT, a \in -1, 1$.

**Table 1  Weight and demand of physical nodes.** The correction and dimension by-dimension strategy adopted by the ULDGA algorithm can adjust the evolutionary direction of genetic particles in time.

| Number | Weight | Requirement |
| --- | --- | --- |
| 1 | 12.29916335 | 12 |
| 2 | 15.01137296 | 18 |
| 3 | 17.42378152 | 2 |
| 4 | 44.84420721 | 7 |
| 5 | 33.82130038 | 11 |
| 6 | 32.61028329 | 20 |
| 7 | 38.06422728 | 20 |
| 8 | 19.13710445 | 2 |
| 9 | 48.24707654 | 2 |
| 10 | 20.900871 | 15 |

(f)  Termination condition judgment: if the termination evolution algebra is reached, the optimal solution will be output, and the algorithm will end. Otherwise, go to the second step to start coding. Also, the flow of the scheme is shown in the figure.

The Metropolis criterion is introduced in the crossover operation, and the excellent individuals in the population are selected to enter the simulated annealing algorithm for annealing optimization. In this way, not only can the superb solution generated in the genetic operation process be retained, but the poor solution can also be accepted with a certain probability.

## THE ANALYSIS OF THE SIMULATION AND RESULT

The simulation experiment is carried out in this chapter on the ULDGA scheme. The platform is Intel 2.3ghz processor, 32GB memory, Windows10 system, matlab r2019c. Select ten distribution points in a particular region and number them 1, 2, …... 10, as shown in Table 1; each customer point carries weight and demand. The Solomon test library contains 56 cases, which are divided into six types of independent data sets with different customer distributions, which are R1 and R2 (random distribution of customer points), C1 and C2 (centralized distribution of customer points), and RC1 and RC2 (distribution characteristics of R class and C class in customer points). The scheduling time of data sets R1, C1 and RC1 is very short, and only a few customers (about 5 to 10) are allowed for each route. In contrast, datasets R2, C2, and RC2 have longer scheduling cycles, allowing the exact vehicle to serve multiple customers (more than 30). It is known that the distribution center is $(12.3, 13, 1)$. The vehicle load for distribution is available ($G= 19.31$). For all the problems in the same type, the customer coordinates are the same, but the width of the time window is different. For example, some customers have tight time windows, which makes it challenging to arrange delivery vehicles. Considering only the first 25 or 50 customers for each problem produces more minor data.

According to Table 1, we have $TabuL = 4$, $Ca = 150$ and $G = 200$. Therefore, this set of parameters is used in this experiment. At the same time, it can be seen from Table 1 that the correction and dimension by-dimension strategy adopted by the ULDGA algorithm

**Table 2  Physical distribution route.** While improving precision, the improved algorithm retains the same performance as different algorithms. Moreover, the specific distribution route is shown in Table 2.

| Vehicle number | Distribution route |
|---|---|
| 1 | 0-12-13-6-7-0 |
| 2 | 0-14-16-13-5-3-0 |
| 3 | 0-1-3-5-7-11-0 |
| 4 | 0-14-9-12-5-0 |
| 5 | 0-4-5-16-0 |
| 6 | 0-2-4-13-0 |
| 7 | 0-5-6-7-3-13-0 |
| 8 | 0-7-8-3-11-12-0 |
| 9 | 0-5-3-7–0 |
| 10 | 0-6-13-11-7-14-0 |

can adjust the evolutionary direction of genetic particles in time, avoid the interference between dimensions, show better local mining ability. The experimental results show that, for most of the multimodal functions, the proposed algorithm achieves a faster convergence effect than other algorithms due to its property. While improving precision, the improved algorithm retains the same performance as different algorithms. Moreover, the specific distribution route is shown in Table 2.

Considering Fig. 4, the improved genetic algorithm determines a longer total path length than the known optimal path length. The x-coordinate is the number of iterations, and the y-coordinate is the minimum value of the objective function obtained by the algorithm. In addition, the relative error of more than 50% of the R1 data set is more significant than 5%. The possible reason is that the distribution of customers for the R1 problem is more uniform. The time window width of each customer point is different, which leads to an increase in the number of vehicles that need to participate in the distribution. Therefore, compared with the current optimal solution, the solution to the R1 problem needs further improvement. For the R2 case, the path length calculated by the improved algorithm is better than the known optimal path length, increasing the number of vehicles used.

In this article, the Intelligent Transportation System (ITS) (*Njoku et al., 2023*) is used to compare with the proposed scheme. ITS improves transportation efficiency, relieves traffic congestion, improves the passing capacity of road network, reduces traffic accidents, reduces energy consumption and reduces environmental pollution through harmonious and close cooperation among people, vehicles and roads. The traditional logistics transportation based on ITS can effectively shorten the in-transit time and improve transportation efficiency through communication and Internet of things technology. The ULDGA scheme and its scheme generate the convergence graph after iteration, as shown in 5. After running the algorithm many times, we get similar calculations because of finding a better solution. The fitness value of the algorithm tends to converge with the increase of iteration times, which shows that the near-optimal solution is obtained. A first-stage iteration curve drops quickly at the beginning, as shown in the figure. After selection, mutation,

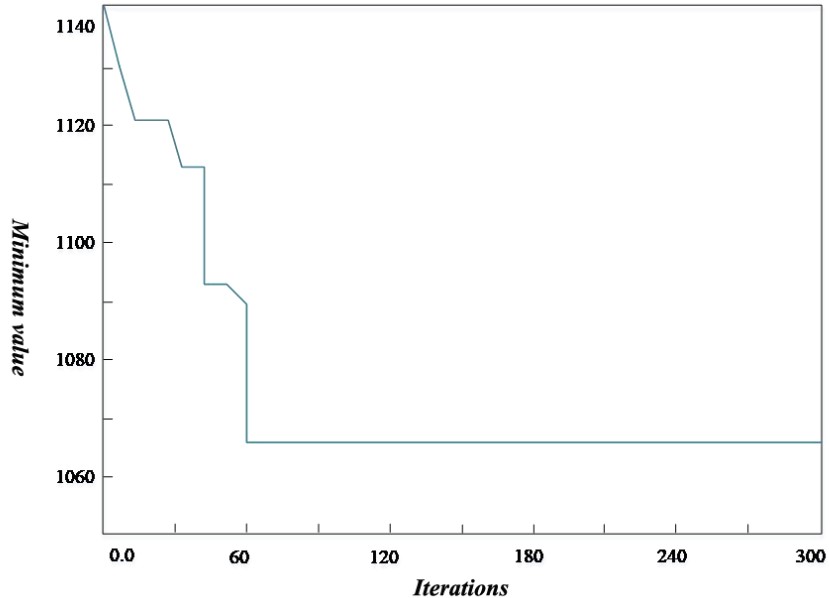

**Figure 4  Iterative circuit diagram.** The improved genetic algorithm determines a longer total path length than the known optimal path length. The $x$-coordinate is the number of iterations, and the $y$-coordinate is the minimum value of the objective function obtained by the algorithm.

crossover, and other optimization operations of the genetic algorithm, the population's fitness is significantly enhanced, and the curve in the graph gradually tends to be flat. It is considered that the optimal solution has been obtained. Through the analysis of Fig. 5, for all 20 examples, the ULDGA not only converges faster but also the relative error between the calculation results of 89.31% of the cases and the current optimal solution is about 1.3%. Due to the problem scale becoming more extensive, the deviation of the results of the last five examples is more than 5%.

Finally, the total cost of the ULDGA scheme is reduced by 55.13% compared with a different scheme. According to the calculation, the transportation mileage of the ULDGA scheme is reduced by 33.17% compared with its scheme. The carbon emissions of the two schemes will not change much. The logistics cost and time penalty cost of the ULDGA scheme are significantly higher than its scheme. Because the logistics times and time of pure electric logistics vehicles significantly impact the total delivery time, it is difficult to respond to the customer's time demand in the delivery task implementation, so it needs to pay more penalty costs. In addition, due to multiple trips to logistics stations, the logistics path of ULDGA is longer, leading to a longer total mileage than its scheme. The above analysis shows that the cost difference between the two schemes is mainly reflected in the difference in logistics cost and time penalty cost. The energy consumption coefficient of scheme 1 is a constant, which remains unchanged with the reduction of vehicle load, so the calculated energy consumption is higher. In addition, more logistics times increase logistics costs and time penalty costs. Compared with the other schemes, the ULDGA scheme has less mileage, less logistics demand in the distribution process, and lower logistics cost and time penalty

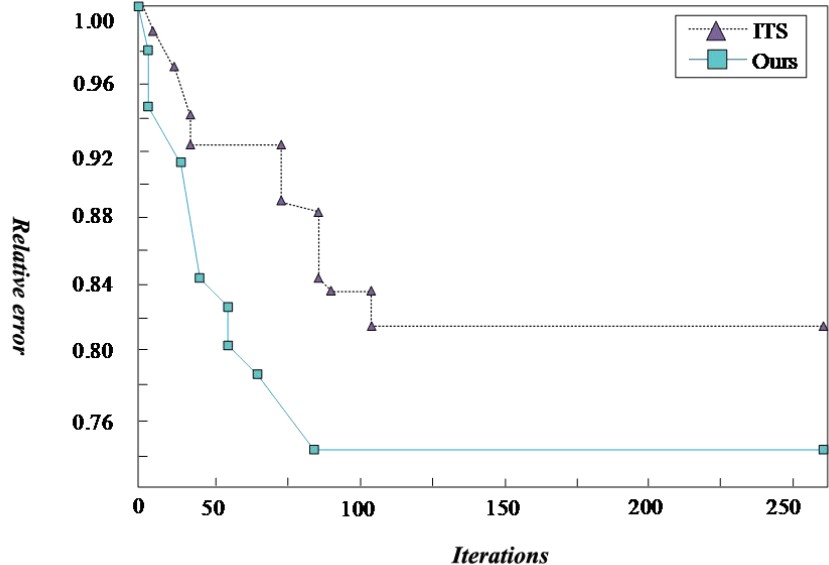

**Figure 5** **Convergence graph.** The ULDGA scheme and its scheme generate the convergence graph after iteration. After running the algorithm many times, we get similar calculations because of finding a better solution.

**Table 3** **Algorithm performance comparison.**

|  | Average cost per yuan | Average time spent/s | The number of times falling into local optimality |
|---|---|---|---|
| ACO | 15272.2 | 16.3 | 10 |
| ABC | 15968.0 | 19.5 | 2 |
| PSO | 12334.6 | 28.5 | 0 |
| Ours | 8234.8 | 25.4 | 1 |

costs. The ULDGA scheme is conducive to vehicle routing planning in enterprise logistics distribution, which shows that the ULDGA scheme is feasible and effective for LPP.

To validate the efficacy of the proposed model, a comparative study is conducted with the Ant Colony Optimization (ACO), Artificial Bee Colony (ABC), and Particle Swarm Optimization (PSO) algorithms in the presented case, as shown in Tables 3 and 4.

Tables 3 and 4 clearly indicates that the proposed model outperforms Ant Colony, ABC, and PSO algorithms in terms of solution quality and stability. This can be attributed to the hybrid genetic algorithm that integrates the strengths of both the basic genetic algorithm and the simulated annealing algorithm, providing global optimization capabilities and effective local optimization, respectively. This allows the hybrid genetic algorithm to overcome local optimization challenges and yield superior results. The introduction of the simulated annealing concept adds an element of randomness that prevents the algorithm from being stuck in local optima, resulting in improved computational outcomes.

**Table 4  Comparison of our method with other method.** This allows the hybrid genetic algorithm to overcome local optimization challenges and yield superior results. The introduction of the simulated annealing concept adds an element of randomness that prevents the algorithm from being stuck in local optima, resulting in improved computational outcomes.

| G | D | Avg. vehicle number | | | Avg total distance | | | Avg insertion time | | | Ratio refuse service | | |
|---|---|---|---|---|---|---|---|---|---|---|---|---|---|
| | | ABC | PSO | Ours | ABC | PSO | Ours | ABC | PSO | Ours | ABC | PSO | Ours |
| | 90 | 14.98 | 15.23 | 15.11 | 1234.98 | 1334.78 | 1223.89 | 5.77 | 18 | 4.32 | 3.22 | 2.15 | 1.35 |
| R1 | 50 | 14.78 | 13.78 | 12.89 | 1234.56 | 1341.52 | 1191 | 19.89 | 20.22 | 18.78 | 2.34 | 1.89 | 1.22 |
| | 10 | 17.27 | 14.17 | 12.89 | 1234.57 | 1549 | 1231.89 | 23.89 | 26.90 | 19.78 | 0.31 | 0.17 | 0.89 |
| | 90 | 12.38 | 12.48 | 12.12 | 909 | 1078.90 | 956.78 | 4.78 | 5.78 | 4.22 | 0.23 | 0.32 | 0.12 |
| C1 | 50 | 11.32 | 11.23 | 10.28 | 872.89 | 789.34 | 897.78 | 5.31 | 5.78 | 5.66 | 0.22 | 0.34 | 0.21 |
| | 10 | 11.23 | 14.22 | 10.43 | 798.34 | 878.23 | 892.71 | 16.23 | 14.54 | 13.54 | 0.12 | 0.43 | 0.32 |
| | 90 | 14.43 | 14.43 | 14.12 | 1487.23 | 1512.12 | 1490.13 | 4.43 | 4.42 | 5.21 | 0.34 | 0.43 | 0.19 |
| RC1 | 50 | 13.34 | 13.54 | 12.89 | 1432.12 | 1419.31 | 1390.45 | 5.31 | 9.31 | 5.23 | 0.12 | 0.45 | 0 |
| | 10 | 12.43 | 12.54 | 12.23 | 1389.12 | 1436.23 | 1378.87 | 4.43 | 4.12 | 3.43 | 0.11 | 0.23 | 0.05 |
| | 90 | 5.23 | 4.12 | 3.12 | 990.12 | 1022 | 976.45 | 6.56 | 12.33 | 5.65 | 0.00 | 0.00 | 0.00 |
| R2 | 50 | 4.24 | 3.21 | 2.89 | 960.29 | 1011.23 | 898 | 5.13 | 7.34 | 4.12 | 0.00 | 0.00 | 0.00 |
| | 10 | 3.89 | 3.12 | 2.67 | 938.06 | 978.22 | 867.12 | 4 | 5.13 | 3.45 | 0.00 | 0.00 | 0.00 |
| | 90 | 4.12 | 3.16 | 3.01 | 678.44 | 623.12 | 679.12 | 3.44 | 5.23 | 4.32 | 0.00 | 0.00 | 0.00 |
| C2 | 50 | 3.34 | 3.11 | 2.87 | 656.31 | 604.22 | 623.10 | 4.12 | 4.22 | 3.98 | 0.00 | 0.00 | 0.00 |
| | 10 | 3.19 | 2.99 | 2.67 | 623.98 | 598.23 | 600.23 | 2.32 | 3.14 | 3.56 | 0.00 | 0.00 | 0.00 |
| | 90 | 6.89 | 5.78 | 5.03 | 1332.11 | 1287 | 1276.89 | 5.67 | 5 | 5.78 | 0.00 | 0.00 | 0.00 |
| RC2 | 50 | 5.78 | 4.29 | 4.21 | 1231.12 | 1178.67 | 1166.33 | 4.23 | 4.16 | 5 | 0.00 | 0.00 | 0.00 |
| | 10 | 4.55 | 4.12 | 4.00 | 1146.24 | 1105.89 | 1132.53 | 3.21 | 3.12 | 4.12 | 0.00 | 0.00 | 0.00 |
| Avg | | 9.08 | 8.64 | 7.97 | 1063.85 | 1097.16 | 1048.48 | 7.17 | 8.86 | 6.67 | 0.39 | 0.36 | 0.24 |

## CONCLUSION

Urban distribution is a significant aspect of the logistics industry, which spans all stages of the logistics chain and has become increasingly vital. This article proposes a hybrid genetic algorithm that incorporates the local optimization capacity of the simulated annealing algorithm and introduces the Metropolis criterion into the cross operation of the basic genetic algorithm. This not only preserves good individuals in the population but also accepts bad chromosomes with a certain probability. Compared to the other two algorithms, the hybrid genetic algorithm is a viable approach to solving the vehicle routing problem by the 8234.8 yuan and 25.4s cost. Furthermore, a novel crossover and mutation strategy is designed based on the existing adaptive algorithms. An evaluation index is also introduced to reflect the population's degree of difference, allowing for the adaptive adjustment of crossover and mutation probabilities. Although the solution quality is optimized, the computation of crossover and mutation operators takes time, and therefore, further research and improvement are required to improve the algorithm's performance.

### Funding
This work was supported by the General Research Project of Humanities and Social Sciences in the colleges and universities of Henan Province, project number 2023-ZDJH-039. The funders had no role in study design, data collection and analysis, decision to publish, or preparation of the manuscript.

### Grant Disclosures
The following grant information was disclosed by the authors:
General research project of Humanities and Social Sciences in the colleges and universities of Henan Province: 2023-ZDJH-039.

### Competing Interests
The authors declare there are no competing interests.

### Author Contributions
- Limin Ran conceived and designed the experiments, performed the experiments, analyzed the data, prepared figures and/or tables, authored or reviewed drafts of the article, and approved the final draft.
- Shengnan Ran conceived and designed the experiments, performed the experiments, analyzed the data, performed the computation work, prepared figures and/or tables, and approved the final draft.
- Chunmei Meng conceived and designed the experiments, performed the experiments, analyzed the data, performed the computation work, authored or reviewed drafts of the article, and approved the final draft.

### Data Availability
The code is available in the Supplementary File. The data is available at Github and Zenodo: https://github.com/TangZwei/OpenTraj/tree/master/docs/data.
None. (2023). Dataset [Data set]. Zenodo. https://doi.org/10.5281/zenodo.7777342.

### Supplemental Information
Supplemental information for this article can be found online at http://dx.doi.org/10.7717/peerj-cs.1347#supplemental-information.

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
