# Peer review of "Green city logistics path planning and design based on genetic algorithm"

_PeerJ Computer Science, doi:10.7717/peerj-cs.1347_

## Round 0.1 · original submission · Major Revisions

The reviewers would like to see some revisions made to your manuscript before publication. Therefore, I invite you to respond to the reviewers' comments and revise carefully your manuscript.

Reviewer 1 ·

Basic reporting

This paper proposes to handle logistics distribution research difficulties, including the delivery of low-performance time orders, the reverse distribution scenario of returned items, and others. The authors used the genetic algorithm (GA) for logistics path planning (LPP) and an enhanced program for LPP, dubbed ULDGA, the overall paper is well written and easy to follow, however, I have following minor concerns to be considered.

Experimental design

The embedding of the Metropolis criterion in the model remains to be further discussed. Total distribution cost includes fixed cost, driving cost, charging cost, penalty cost and carbon emission cost. Is it reflected in the Metropolitan Standards?

The limitations of the research and the future development direction need further elaboration and analysis. The conclusion part needs to adjust the language expression, and the elaboration of this part is too lengthy. The author needs to simplify this part.

Validity of the findings

In the result analysis, the author compares the ULDGA model with ITS model, but there is a lack of a corresponding introduction to the characteristics of ITS, which is not conducive to highlighting the advantages of the proposed model

Reviewer 2 ·

Basic reporting

Based on the background of logistics distribution, this paper adopts genetic algorithm for logistics route planning, and designs an improved genetic algorithm scheme with the Metropolis criterion. In order to overcome the deficiency of the local search ability of the genetic algorithm, the simulated annealing algorithm with strong local search ability is mixed into the genetic algorithm. An improved genetic algorithm with Metropolis criterion is designed, which combines annealing operation with genetic operation, retains the best chromosome in each generation population, and accepts inferior chromosome with certain probability to increase the probability of searching out the local optimal solution, so as to achieve global optimization. This research is helpful to design a reasonable logistics distribution route, so as to strengthen the core competitiveness of enterprises. Thank you for submitting and for your patience. We are working through submissions in the order they are submitted. Please address them all before resubmitting.
(1) Uncommon abbreviations should be spelled out at first use. Do not include footnotes or references.
(2) The end of the introduction should make a supplementary analysis of the research structure.
(3) In the description of Section 3, two assumptions are put forward to limit the constraints. This part of the description is rather confused, lack of a certain logical relationship.
(4) Cost should be the core factor affecting logistics distribution. In the construction of the model, I suggest that the author add distribution cost as one of the constraint factors of the model, instead of only establishing a model with path, electricity, time and load as constraints.
(5) In the algorithm design, there is not enough connection between the codes of different factors, and the distribution center codes of the starting point and the ending point are not discussed. This part needs to be improved;
(6) At present, the results are not enough to support the author's view, and there is no evidence that the algorithm can solve the problem of low convergence accuracy in high-dimensional environment.
(7) In the designed model, what is the specific arrangement and updating mode of chromosome sequence? If necessary, an example of a path planning problem can be used to better illustrate the author's point of view.

Experimental design

Based on the background of logistics distribution, this paper adopts genetic algorithm for logistics route planning, and designs an improved genetic algorithm scheme with the Metropolis criterion. In order to overcome the deficiency of the local search ability of the genetic algorithm, the simulated annealing algorithm with strong local search ability is mixed into the genetic algorithm. An improved genetic algorithm with Metropolis criterion is designed, which combines annealing operation with genetic operation, retains the best chromosome in each generation population, and accepts inferior chromosome with certain probability to increase the probability of searching out the local optimal solution, so as to achieve global optimization. This research is helpful to design a reasonable logistics distribution route, so as to strengthen the core competitiveness of enterprises. Thank you for submitting and for your patience. We are working through submissions in the order they are submitted. Please address them all before resubmitting.
(1) Uncommon abbreviations should be spelled out at first use. Do not include footnotes or references.
(2) The end of the introduction should make a supplementary analysis of the research structure.
(3) In the description of Section 3, two assumptions are put forward to limit the constraints. This part of the description is rather confused, lack of a certain logical relationship.
(4) Cost should be the core factor affecting logistics distribution. In the construction of the model, I suggest that the author add distribution cost as one of the constraint factors of the model, instead of only establishing a model with path, electricity, time and load as constraints.
(5) In the algorithm design, there is not enough connection between the codes of different factors, and the distribution center codes of the starting point and the ending point are not discussed. This part needs to be improved;
(6) At present, the results are not enough to support the author's view, and there is no evidence that the algorithm can solve the problem of low convergence accuracy in high-dimensional environment.
(7) In the designed model, what is the specific arrangement and updating mode of chromosome sequence? If necessary, an example of a path planning problem can be used to better illustrate the author's point of view.

Validity of the findings

Based on the background of logistics distribution, this paper adopts genetic algorithm for logistics route planning, and designs an improved genetic algorithm scheme with the Metropolis criterion. In order to overcome the deficiency of the local search ability of the genetic algorithm, the simulated annealing algorithm with strong local search ability is mixed into the genetic algorithm. An improved genetic algorithm with Metropolis criterion is designed, which combines annealing operation with genetic operation, retains the best chromosome in each generation population, and accepts inferior chromosome with certain probability to increase the probability of searching out the local optimal solution, so as to achieve global optimization. This research is helpful to design a reasonable logistics distribution route, so as to strengthen the core competitiveness of enterprises. Thank you for submitting and for your patience. We are working through submissions in the order they are submitted. Please address them all before resubmitting.
(1) Uncommon abbreviations should be spelled out at first use. Do not include footnotes or references.
(2) The end of the introduction should make a supplementary analysis of the research structure.
(3) In the description of Section 3, two assumptions are put forward to limit the constraints. This part of the description is rather confused, lack of a certain logical relationship.
(4) Cost should be the core factor affecting logistics distribution. In the construction of the model, I suggest that the author add distribution cost as one of the constraint factors of the model, instead of only establishing a model with path, electricity, time and load as constraints.
(5) In the algorithm design, there is not enough connection between the codes of different factors, and the distribution center codes of the starting point and the ending point are not discussed. This part needs to be improved;
(6) At present, the results are not enough to support the author's view, and there is no evidence that the algorithm can solve the problem of low convergence accuracy in high-dimensional environment.
(7) In the designed model, what is the specific arrangement and updating mode of chromosome sequence? If necessary, an example of a path planning problem can be used to better illustrate the author's point of view.

Reviewer 3 ·

Basic reporting

Aiming at the weakness of GA algorithm's local search ability, the author combines SA algorithm with SA algorithm, which has strong local search ability, and introduces Metropolis criterion into the cross operation of genetic algorithm, which not only retains good individuals in the population, but also accepts inferior chromosomes with a certain probability. This algorithm is used to solve the vehicle routing problem with time window. With the rapid development of urban logistics industry, urban distribution, as an important form of logistics industry, runs through every stage of logistics links and gradually becomes the focus of the logistics industry. This work is helpful to solve the problems of high logistics operation cost, waste of transport capacity and low level of distribution service in the development of urban logistics. This paper has some innovation, but it needs to be modified.

Why the author divides user data into six different independent data sets, including random distribution of customer points, centralized distribution of customer points and the combination of the two; .
In the author's opinion, according to the different treatment of time window constraints, they can be divided into two categories, but in the subsequent description, they are divided into four points, which need to be explained and modified accordingly;
In addition, it also introduces the research methods of logistics vehicle route planning in the introduction, such as "integer programming" and "mixed integer programming". However, this part of the content is repeated with Section 2, so it is suggested that the author should revise it carefully.

Experimental design

No Comments

Validity of the findings

No Comments

Additional comments

Aiming at the weakness of GA algorithm's local search ability, the author combines SA algorithm with SA algorithm, which has strong local search ability, and introduces Metropolis criterion into the cross operation of genetic algorithm, which not only retains good individuals in the population, but also accepts inferior chromosomes with a certain probability. This algorithm is used to solve the vehicle routing problem with time window. With the rapid development of urban logistics industry, urban distribution, as an important form of logistics industry, runs through every stage of logistics links and gradually becomes the focus of the logistics industry. This work is helpful to solve the problems of high logistics operation cost, waste of transport capacity and low level of distribution service in the development of urban logistics. This paper has some innovation, but it needs to be modified.

Why the author divides user data into six different independent data sets, including random distribution of customer points, centralized distribution of customer points and the combination of the two; .
In the author's opinion, according to the different treatment of time window constraints, they can be divided into two categories, but in the subsequent description, they are divided into four points, which need to be explained and modified accordingly;
In addition, it also introduces the research methods of logistics vehicle route planning in the introduction, such as "integer programming" and "mixed integer programming". However, this part of the content is repeated with Section 2, so it is suggested that the author should revise it carefully.

---

## Round 0.2 · Major Revisions

The revised article was examined, and it was concluded that the following corrections should be made. In particular, I am of the opinion that the article should be completely reconsidered in terms of language and innovation and so the corrections should be made accordingly.

1- Abbreviations should not be used in the abstract.
2- The content of the study in the abstract is not emphasized enough. The abstract should be rewritten as clear and rich.
3- Literature analysis is not sufficient in the introduction section. The literature given in the references is insufficient. The originality of the study and its contribution to the literature have not been adequately addressed.
4- Figure 4 needs to be explained.
5- In Figure 5, it should be preferred to use words such as "developed method" instead of the word "ours".
6- For Figure 5, it is necessary to choose words suitable for the literature, such as the convergence graph.
7- ULDGA must be compared with Ant colony, ABC and PSO algorithms and the compared results must be updated and analyzed accordingly. Authors need to focus more on obtaining and interpreting comparison results.
8- "Conclusion" section is insufficient, it should be rewritten.
9- In section "Acknowledgments", the sentence "The authors would like to thank anonymous reviewers who contributed valuable comments to this article" should be removed.
10. Professional help should be sought in terms of English and article writing technique.
11. I'm unsure of the contribution of this work overall - the motivation of the article should be constructed according to this comment. Please make it clear.

Reviewer 1 ·

Basic reporting

All changes have been done

Experimental design

All changes have been done

Validity of the findings

All changes have been done

Reviewer 2 ·

Basic reporting

This section is very good.

Experimental design

This section is very good.

Validity of the findings

This section is very good.

Additional comments

no comment. It is ok.

Reviewer 3 ·

Basic reporting

May be accepted in the present form

Experimental design

May be accepted in the present form

Validity of the findings

May be accepted in the present form.

Additional comments

Accepted

---

## Round 0.3 · Major Revisions

I acknowledge that writing an article involves technical difficulties and is part of scientific study and education. That's why I patiently make some recommendations for the article to gain a scientific identity. There has been some progress in the revision. However, additional corrections are needed. Requested corrections are attached as pdf.

---

## Round 0.4 · accepted · Accept

In the article, corrections were made by the authors as according to the reviewers' comments.